# Identification of Morphologic Criteria Associated with Biochemical Recurrence in Intraductal Carcinoma of the Prostate

**DOI:** 10.3390/cancers13246243

**Published:** 2021-12-13

**Authors:** Mame-Kany Diop, Roula Albadine, André Kougioumoutzakis, Nathalie Delvoye, Hélène Hovington, Alain Bergeron, Yves Fradet, Fred Saad, Dominique Trudel

**Affiliations:** 1Centre de recherche du Centre hospitalier de l’Université de Montréal (axe Cancer) and Institut du cancer de Montréal, 900 Saint-Denis, Montréal, QC H2X 0A9, Canada; mame-kany.diop.chum@ssss.gouv.qc.ca (M.-K.D.); nathalie.delvoye.chum@ssss.gouv.qc.ca (N.D.); fred.saad@umontreal.ca (F.S.); 2Department of Pathology and Cellular Biology, Université de Montréal, 2900 Boulevard Édouard-Montpetit, Montreal, QC H3T 1J4, Canada; roula.albadine.med@ssss.gouv.qc.ca; 3Department of Pathology, Centre hospitalier de l’Université de Montréal, 1051 Sanguinet, Montréal, QC H2X 0C1, Canada; andre.kougioumoutzakis.med@ssss.gouv.qc.ca; 4Laboratoire d’Uro-Oncologie Expérimentale, Centre de recherche du CHU de Québec-Université Laval (axe Oncologie), Hôpital L’Hôtel-Dieu de Québec, 10 McMahon, Québec City, QC G1R 3S1, Canada; helene.hovington@crchudequebec.ulaval.ca (H.H.); alain.bergeron@crchudequebec.ulaval.ca (A.B.); yves.fradet@crchudequebec.ulaval.ca (Y.F.); 5Department of Surgery, Université Laval, 2325 rue de l’Université, Québec City, QC G1V 0A6, Canada; 6Department of Urology, Centre hospitalier de l’Université de Montréal, 1051 Sanguinet, Montréal, QC H2X 0C1, Canada

**Keywords:** prostate cancer, intraductal carcinoma of the prostate, morphologic criteria, biochemical recurrence, prostate cancer prognosis, radical prostatectomy

## Abstract

**Simple Summary:**

Despite the strong association of the aggressive intraductal carcinoma of the prostate (IDC-P) with an increased risk of biochemical recurrence (BCR), around 40% of men remain BCR-free five years post-surgery. In this retrospective study, we aimed to evaluate the prognostic value of several morphological criteria of IDC-P using BCR as the endpoint. In multivariate analysis (validation cohort, *n* = 69), the presence of cells with irregular nuclear contours (CINC) or blood vessels was independently associated with an increased risk of BCR (hazard ratio [HR] 2.32, 95% confidence interval [CI] 1.09–4.96, *p* = 0.029). Furthermore, when combining the criteria, the presence of any CINC, blood vessels, high mitotic score, or comedonecrosis showed a stronger association with BCR (HR 2.74, 95% CI 1.21–6.19, *p* = 0.015). Provided that our findings are validated in larger cohorts, evaluation of morphologic features of IDC-P could serve as a risk stratification tool for patients with IDC-P.

**Abstract:**

Intraductal carcinoma of the prostate (IDC-P) is an aggressive subtype of prostate cancer strongly associated with an increased risk of biochemical recurrence (BCR). However, approximately 40% of men with IDC-P remain BCR-free five years after radical prostatectomy. In this retrospective multicenter study, we aimed to identify histologic criteria associated with BCR for IDC-P lesions. A total of 108 first-line radical prostatectomy specimens were reviewed. In our test cohort (*n* = 39), presence of larger duct size (>573 µm in diameter), cells with irregular nuclear contours (CINC) (≥5 CINC in two distinct high-power fields), high mitotic score (>1.81 mitoses/mm^2^), blood vessels, and comedonecrosis were associated with early BCR (<18 months) (*p* < 0.05). In our validation cohort (*n* = 69), the presence of CINC or blood vessels was independently associated with an increased risk of BCR (hazard ratio [HR] 2.32, 95% confidence interval [CI] 1.09–4.96, *p* = 0.029). When combining the criteria, the presence of any CINC, blood vessels, high mitotic score, or comedonecrosis showed a stronger association with BCR (HR 2.74, 95% CI 1.21–6.19, *p* = 0.015). Our results suggest that IDC-P can be classified as low versus high-risk of BCR. The defined morphologic criteria can be easily assessed and should be integrated for clinical application following validation in larger cohorts.

## 1. Introduction

Intraductal carcinoma of the prostate (IDC-P) is now accepted as a biologically distinct entity of prostate cancer (PCa) [1]. Since it was first described as a “ductal spread in prostatic carcinoma” by Kobi et al. in 1985 [2], multiple studies clarified the concept of IDC-P as a lumen-spanning spread and growth of neoplastic epithelial cells in ducts retaining basal cells [3,4,5,6] and almost always found adjacent to invasive, mostly aggressive, PCa [7,8,9,10,11,12,13]. Indeed, IDC-P without an adjacent invasive component was observed in only 0.06% [9] to 2.6% [10] of prostate biopsies. The incidence of IDC-P varied from 2.1% in patients with low-risk PCa to 56.0% in patients with metastatic or recurrent disease [14].

The presence of basal cells is the main feature that distinguishes IDC-P from high-grade invasive carcinoma, whether cribriform, solid, or with comedonecrosis. IDC-P also shares some features with high-grade prostatic intraepithelial neoplasia, but the architectural and cytologic atypia found in IDC-P can discriminate between the two types of lesions. However, the presence of borderline morphology can sometimes render the diagnosis more subjective [7,15,16], and several sets of morphologic criteria have been proposed to detect IDC-P [4,6,9], complicating its identification [17,18]. The most frequently used criteria, established by Guo and Epstein, are highly specific and designed to identify prostates with IDC-P, but without associated invasive carcinoma [9]. 

Regardless of the diagnostic criteria, IDC-P has been associated with adverse pathological features including larger tumor volume [19,20], increased stage [6,19], positive surgical margins [6], and higher grade [2,6,19,20]. Moreover, IDC-P was shown as an independent predictor of progression-free survival and biochemical recurrence (BCR) after radical prostatectomy (RP) or hormone therapy [9,21,22]. However, approximately 40% of patients harboring IDC-P are still BCR-free after five years of follow-up [23]. 

Given the large morphological spectrum of IDC-P, we hypothesized that specific features of IDC-P are associated with poor prognosis and can be used to identify patients who will progress more quickly. Thus, the aim of this study was to find adverse morphologic criteria in IDC-P lesions associated with BCR. We identified five morphological criteria associated with early BCR (<18 months) in our test cohort including two that were independently associated with an increased risk of BCR in our validation cohort.

## 2. Materials and Methods

### 2.1. Patients and Ethics

We reviewed RP specimens of the PCa biobank of the Centre hospitalier de l’Université de Montréal (CHUM) (Center 1) collected between 2013 and 2018 and of the URO-1 biobank of the Centre de recherche du CHU de Québec-Université Laval (CHUQc-UL) (Center 2) collected between 1990 and 2003. A retrospective search identified men with first-line RP and IDC-P reported in patient files. Older specimens for which IDC-P was not routinely reported were also added in the study after the review of their RP specimens revealed the presence of IDC-P. All men signed an informed consent form to participate in the biobanks. Investigations were performed after approval by the CHUM Research Ethics Committee (research project MP-02-2018-7450). 

### 2.2. Histologic Evaluation

Hematoxylin and eosin (H&E) or hematoxylin phloxine saffron (HPS) slides (as routinely used for diagnosis in the earlier cases) of RP specimens were first examined by an observer with experience in the identification of IDC-P (M.-K.D., Observer 1), then reviewed by a pathologist expert in IDC-P (D.T.), both blinded to patient progression and clinical information. IDC-P was diagnosed according to criteria proposed by Guo and Epstein [9], which includes an intraductal proliferation of cancer cells forming solid or dense cribriform patterns, or loose cribriform or micropapillary patterns with the following features: oversized nuclei (nuclear size 6× normal), marked pleomorphism, frequent mitotic figures, and frequent comedonecrosis. When diagnosis was uncertain, immunohistochemistry staining for p63 was performed to confirm the presence of basal cells in IDC-P. For each specimen, the slide with the highest amount of IDC-P was selected as the most representative slide, then scanned using a Nanozoomer whole slide scanner (Hamamatsu, Bridgewater, NJ, USA) and analyzed to identify morphologic criteria of the IDC-P lesions. We evaluated the total amount of IDC-P in the RP, architectural (duct size, solid pattern, comedonecrosis), and cytonuclear (nuclei size and shape, pyknotic nuclei, mitosis) criteria, and some others such as the presence of blood vessels. Solid patterns, comedonecrosis, and irregularly shaped nuclei were assessed by recording their presence or absence on each slide. Duct size was calculated by measuring the largest affected duct on each slide, avoiding areas of tangential sectioning. Other criteria such as nuclei size, pyknotic nuclei, and mitosis were semi-quantified in randomly selected 10 high-power fields (400× magnification or 0.2 mm^2^ × 10, or less for smaller lesions) within IDC-P lesions, except for pyknotic nuclei where fields were selected in the immediate adjacent invasive carcinoma. Potential adverse criteria were counted in selected fields, and then quartiles were calculated for each slide. The third quartile of distribution was used as a cut-off point between the “low” and “high” categories. 

### 2.3. Cohorts

RPs were divided into two cohorts: test cohort and validation cohort. The test cohort included eligible men from Center 1, randomly selected to identify morphological criteria associated with poor prognosis. Patients from the test cohort who experienced BCR less than 18 months post-surgery were placed in the early BCR group [24,25], while patients who were BCR-free 18 months post-surgery were placed in the late BCR group. 

Features associated with early BCR in the test cohort were then evaluated in the validation cohort, an independent cohort including randomly selected eligible men from Center 1 and all eligible men from Center 2 with available slides containing IDC-P. H&E or HPS slides were reviewed, and the slide with the highest amount of IDC-P was selected as the most representative slide, scanned, and analyzed, without knowledge of clinical information. 

### 2.4. Clinical Data Collection and Endpoints

Age and clinicopathological characteristics including pathological (p) stage, modified Gleason grading system/Grade Group (GG) grading [1,26], prostate-specific antigen (PSA) serum values, and margin status were collected from patient files. When necessary, a pathologist (D.T. or R.A.) reassessed pathological staging and tumor grading according to the eighth edition of the American Joint Committee on Cancer’s Prostate Staging System and GG grading [1,26]. 

Clinical follow-up data were reviewed for the occurrence of BCR after surgery. The main endpoint was the association between adverse criteria in IDC-P and BCR. BCR was defined as rising PSA > 0.2 ng/mL after RP or an increase in serum PSA that required post-operative treatment. 

### 2.5. Interobserver Agreement

Slides from both cohorts were assessed without knowledge of previous results and of clinical outcome by a pathology resident with experience in uropathology (A.K., Observer 2) using the same criteria as Observer 1 (M.-K.D.). For each slide, CINC, blood vessels, and comedonecrosis were evaluated on all IDC-P lesions while mitosis was evaluated in 10 high-power fields (0.2 mm^2^ × 10, or less for smaller lesions) that were randomly selected in IDC-P.

### 2.6. Statistical Analysis

Statistical analyses were conducted using IBM SPSS Statistics 25 (SPSS Inc.) following the REMARK guidelines [27]. The time to BCR was calculated from the date of RP until the date of BCR or last known PSA date/follow-up date. Univariate methods included Fisher’s exact test, Welch’s test, Pearson’s chi-square test, and the Mann–Whitney U test. BCR-free survival was evaluated using the Kaplan–Meier method, log-rank test, and Cox regression analysis. Univariate and multivariate Cox regression models were used to estimate the hazard ratios (HRs) for adverse criteria. In univariate analysis, we examined four known predictors of BCR: pre-operative PSA level, pT stage, RP GG (grouped as 1–2, 3, and 4–5 to reduce null-groups), and margin status. For multivariate analyses, the GG was included in the model. Interobserver agreement was calculated using Cohen’s kappa (κ) coefficient. Kappa values between 0.00 and 0.20 were interpreted as slight agreement, between 0.21 and 0.40 for fair agreement, between 0.41 and 0.60 for moderate agreement, between 0.61 and 0.80 for substantial agreement, and over 0.80 for excellent agreement [28]. A two-sided *p*-value < 0.05 was considered statistically significant.

## 3. Results

### 3.1. Clinicopathological Characteristics of Patients

The general workflow of the study is illustrated in Figure 1. The test cohort included 45 eligible men from Center 1, randomly selected for a thorough examination of the morphological criteria for IDC-P lesions. The five significant criteria defined by the test cohort were evaluated in the validation cohort, which consisted of 36 patients from Center 1 and 34 patients from Center 2. Seven patients from Center 1 with a postoperative follow-up of less than 18 months were excluded, bringing the final number of patients to 39 in the test cohort and 69 in the validation cohort. 

Clinicopathological characteristics of all 108 patients are summarized in Table 1. Patients from Center 2 underwent RP between 1990 and 2003, which accounted for a longer median follow-up compared to patients from Center 1 who underwent RP between 2013 and 2018 (157 months vs. 47 months, *p* < 0.001). Accordingly, the BCR rate was higher in Center 2, with BCR developing in 59% of patients from Center 2 compared to 25% of patients from Center 1 (*p* = 0.035). However, the prevalence of early BCR was similar in both centers with a mean of 22% of patients developing early BCR (*p* = 1.000). Patients from Center 1 tended to have higher stage cancers with higher grades, as confirmed by higher rates of extraprostatic extension (86% vs. 59%, *p* < 0.001), lymphovascular invasion (49% vs. 18%, *p* = 0.001), and a higher tendency for seminal vesicle invasion (35% vs. 18%, *p* = 0.088). When comparing between the test and validation cohorts, stage (*p* = 0.099), GG (*p* = 0.388), BCR (*p* = 0.653), and median follow-up (*p* = 0.086) did not show statistically significant differences.

### 3.2. IDC-P Feature Descriptions

Amount of IDC-P was evaluated by reporting the total area of IDC-P and the percentage of IDC-P among all cancer (area of IDC-P/area of cancer × 100) in each slide. A threshold of 13.31 mm^2^ and 25% of IDC-P were established according to the third quartile of distribution, resulting in 10 patients (26%) in the “high” categories for area and percentage of IDC-P.

Dilatation of ducts caused by intraductal proliferation of cancer cells was assessed by measuring the transverse diameter of the largest duct in each specimen (Figure 2a). Specimens containing at least one IDC-P lesion with a diameter over 573 μm (third quartile of the distribution) were classified as the “larger duct size” group (*n* = 11; 28%). 

Nuclear contours were deemed irregular when nuclei had tortuous contours or exhibited angles similar to the irregular nuclei observed in papillary thyroid carcinoma [29], in contrast to the typical round nuclear shape (Figure 2b). Since CINC were scarce, at least two distinct high-power fields (0.2 mm^2^) containing at least five CINC were required to categorize patients in the irregular nuclear contour group (*n* = 8; 21%). 

Mitotic activity was evaluated in a semi-quantitative fashion. Mitotic figures (Figure 2c) were counted in 10 high-power fields (0.2 mm^2^) randomly selected within IDC-P lesions. For patients with smaller IDC-P lesions, the maximum number of fields were taken into account when less than 10 fields were available (mean number of fields: 8.2, standard deviation (SD) 2.5; median number of fields: 10, interquartile range (IQR) 6.5–10). A mean number of mitotic figures per mm^2^ was then calculated for each section, and a threshold of 1.81 mitotic figures per mm^2^ was established according to the third quartile of distribution. Patients with more than 1.81 mitotic figures per mm^2^ were placed in the high mitotic score group (*n* = 10; 26%). 

Interestingly, out of the 39 specimens, seven (18%) had blood vessels within IDC-P lesions and were mostly identified by the presence of groups of red blood cells surrounded by endothelial cells in a restricted area of the IDC-P. Blood vessels within IDC-P lesions were not surrounded by stroma; no structures were seen between endothelial cells and the neighboring cancer cells in IDC-P. Blood vessels were recorded as absent or present. Patients were included in the “presence of vessel” category as soon as one blood vessel was identified in any lesion of interest (Figure 2d). 

Similarly, 11 specimens (28%) harbored IDC-P with comedonecrosis, which was defined as central necrotic cells within dense cribriform or solid proliferation in a duct with preservation of basal cells. Necrotic cells had to be clearly visible in the center of the duct for the lesion to be counted as comedonecrosis, which was also recorded as absent or present. Patients were included in the “presence of comedonecrosis” category whenever necrosis was identified within any IDC-P lesion (Figure 2e). Furthermore, 11 specimens (28%) showed IDC-P with a solid pattern, indicating that more than 95% of the lumen of a duct was occupied by cancer cells (see Appendix A). 

For nuclei size, we determined that six times the size of normal nuclei was approximately equal to a nucleus of 180 μm² based on the size of nuclei in benign epithelial luminal cells (see Appendix A). Using the same high-power fields as for the mitotic figures, nuclei were counted and generated a threshold of 0.45 nucleus ≥ 180 μm² per mm^2^ according to the third quartile of distribution. Patients with more than 0.45 nucleus ≥ 180 μm² per mm^2^ were placed in the “high number of nuclei six times the size of normal nuclei” group (*n* = 10; 26%). 

Unlike the other criteria, pyknotic nuclei were assessed in the adjacent invasive cancer (see Appendix A). Ten high-power fields or less with smaller cancers (mean number of fields: 9.3 (SD 1.5); median number of fields: 10 (IQR 9.5–10)) were randomly selected in the adjacent invasive cancer (within less than 1 mm of IDC-P), and each field was assigned a score depending on the percentage of pyknotic nuclei among all cancer cell nuclei: “negative” for less than 5% and “positive” for 5% and more pyknotic nuclei. A mean number of positive fields was then calculated for each specimen, and a threshold of 2.27 positive fields per mm^2^ was established according to the third quartile of distribution, resulting in 10 patients in the “high” category (26%).

### 3.3. Association between Larger Duct Size, CINC, High Mitotic Score, Blood Vessels, Comedonecrosis, and Early BCR in the Test Cohort

Figure 2f illustrates the distribution of five potential criteria among the 39 patients from the test cohort. Adverse criteria were more prevalent in patients who had early BCR (median number of criteria of 0 in the late BCR group vs. four in the early BCR group). Two-sided Fisher’s exact test confirmed that early BCR occurred more frequently in patients with larger duct size (71% in the early BCR group vs. 19% in the late BCR group, *p* = 0.012), CINC (57% in the early BCR group vs. 13% in the late BCR group, *p* = 0.022), high mitotic score (71% in the early BCR group vs. 16% in the late BCR group, *p* = 0.007), vessels (57% in the early BCR group vs. 9% in the late BCR group, *p* = 0.012), and comedonecrosis (71% in the early BCR group vs. 19% in the late BCR group, *p* = 0.012). Volume of IDC-P (area), percentage of IDC-P, presence of solid patterns, large nuclei, and pyknotic nuclei did not show a significant association with BCR.

### 3.4. Validation of Time to BCR According to Identified Adverse Criteria 

The median follow-up of patients from the validation cohort was 67 months (IQR: 32–157). During follow-up, 31 men experienced BCR. Kaplan–Meier curves of the five proposed criteria are presented in Figure 3. The median time to BCR was significantly shorter in patients with CINC (61 months, 95% confidence interval [CI] 3.6–120 vs. median survival not yet reached in patients without CINC; *p* = 0.009) and in patients with vessels in IDC-P (39 months, 95% CI 0–107.7 vs. 96 months, 95% CI not calculable in patients without vessels in IDC-P; *p* = 0.022). High mitotic score and comedonecrosis were not statistically associated with BCR (*p* = 0.556 and *p* = 0.163, respectively) and no significant difference was seen when stratifying patients according to duct size (*p* = 0.914). A univariate Cox proportional hazard analysis confirmed these results (Table 2). Only the presence of CINC and vessels were significantly associated with shorter BCR-free survival (HR 2.60, 95% CI 1.24–5.47, *p* = 0.012; and HR 2.24, 95% CI 1.10–4.56, *p* = 0.026, respectively). In addition, higher GG and more advanced pT stage showed statistically significant association with poor BCR-free survival (HR 2.68, 95% CI 1.03–7.03, *p* = 0.045 between stages pT2 and pT3b; HR 2.72, 95% CI 1.07–6.92, *p* = 0.036 between GG 1–2 and 3; HR 5.049, 95% CI 1.96–13.02, *p* = 0.001 between GG 3 and 4–5).

We then assessed the potential of combining the two criteria associated with an increased risk of BCR: presence of CINC and vessels. Interestingly, both Kaplan–Meier curves (Figure 4a) and univariate analysis (Table 3) revealed that the presence of a single adverse criterion rather than the number of criteria had the biggest impact on BCR-free survival. Therefore, we compared survival curves when stratifying for absence or presence of at least one criterion (Figure 4b). A median survival of 62 months was obtained compared to median survival not yet reached in patients without any criteria (*p* = 0.002). A multivariate Cox proportional hazard analysis was performed to evaluate prognostic value of the presence of any CINC or vessels while controlling for GG (Table 3). The presence of criteria remained significantly associated with an increased risk of BCR (HR 2.32, 95% CI 1.09–4.96, *p* = 0.029). Since high mitotic score and comedonecrosis showed a trend toward shorter BCR-free survival, we also tested the combination of three to four criteria: CINC, vessels, and high mitotic score; CINC, vessels and comedonecrosis; or CINC, vessels, high mitotic score, and comedonecrosis (see Appendix A). The presence of any criteria retained its prognostic value in all combinations (HR 2.64, 95% CI 1.18–5.89, *p* = 0.018; HR 2.70, 95% CI 1.23–5.91, *p* = 0.013; and HR 2.74, 95% CI 1.21–6.19, *p* = 0.015, respectively). Interestingly, when examining the distribution of criteria in the validation cohort, regardless of the combination, most men with adverse criteria only had one adverse criterion (range 76–86%), and the presence of vessels was the predominant criterion (see Appendix A). 

The interobserver agreement was moderate for mitotic score, comedonecrosis, and vessel interpretation (κ 0.60, 95% CI 0.40–0.80; κ 0.51, 95% CI 0.33–0.68; and κ 0.47, 95% CI 0.28–0.66, respectively, *p* < 0.001) and fair for CINC interpretation (κ 0.33, 95% CI 0.14–0.51, *p* < 0.001) (Table 4).

In addition, we evaluated the association of each proposed adverse criteria of IDC-P with GG, pT stage, and surgical margin status in the test and the validation cohorts (see Appendix A). In the test cohort, the presence of CINC (*p* = 0.035), vessels (*p* = 0.027), and high mitotic scores (*p* = 0.044) were associated with higher grades and only the presence of high mitotic scores was associated with higher stage cancers (*p* = 0.029). However, no significant association were observed in the validation cohort.

## 4. Discussion

IDC-P has been consistently linked to high-grade PCa disease and poor prognosis [30], however, a significant portion of patients with IDC-P show a slow disease progression [23]. Recent studies have focused on distinguishing IDC-P from other lesions [16,31,32]. In this multicenter study, we were the first to evaluate the prognostic value of several histologic criteria in IDC-P. We focused on the morphological diversity between IDC-P lesions and found five morphologic criteria associated with early BCR in our test cohort: the presence of larger duct size, high mitotic score, comedonecrosis, vessels, or CINC. The presence of CINC and vessels, alone and together, remained significantly associated with BCR in the validation cohort. 

Few studies have investigated the link between the morphology of IDC-P and clinical outcome. In a study by Wilcox et al., only 20% of men with solid and comedonecrosis IDC-P (*n* = 28) were progression-free after five years of follow-up compared to 65% of men with cribriform IDC-P (*n* = 80) [19]. Cohen et al. found that 67% of men (12/18) with solid and cribriform IDC-P relapsed after three years of follow-up compared to roughly more than 15% of men (1/6) with trabecular IDC-P [3]. Here, we examined several histologic criteria including some that had not previously been evaluated to predict the clinical outcomes of patients with IDC-P such as CINC and the presence of blood vessels.

Among our evaluated criteria, comedonecrosis, mitosis, and duct size have been previously linked to IDC-P. Recently, three research groups confirmed an intraductal component of comedonecrosis by finding retaining basal cells for all or part of comedonecrosis foci in 78% to 95% of PCa cases with comedonecrosis [31,33,34]. Moreover, among the 27 prostate biopsies with only IDC-P, Guo and Epstein reported mitoses in 20 cases (74%) [9]. In addition to comedonecrosis (22/27) and marked pleomorphism (18/27), mitoses were the most frequently observed cytological feature in IDC-P. For duct size, Cohen et al. proposed that ducts must be enlarged to at least twice the size of adjacent benign glands to be IDC-P [4]. To our knowledge, no study has assessed vessels and CINC in IDC-P lesions. Although the presence of vessels has not been previously reported inside ducts colonized by cancer cells, this feature was observed quite frequently in our cohort (*n* = 7 in the test cohort and *n* = 20 in the validation cohort; total of 25%). Interestingly, IDC-P has been associated with hypoxia of the prostate [35]. Regarding CINC, they are more frequent in high-grade PCa tumors [36,37] and may reflect the lack of nuclear differentiation in more aggressive cancers. In our study, unlike what has been reported [3,19], IDC-P with a solid pattern was not associated with an increased risk of BCR. Since a mixture of architectural intraductal patterns can coexist within the same tumor [3,19], the presence of a solid pattern could have been missed by reviewing a single slide per patient. Of note, we defined solid epithelial masses forming a solid intraductal pattern as cancer cells occupying more than 95% of the lumen of a duct, but used definitions were not provided in previous studies [3,6,19]. Additionally, the low number of patients with adverse criteria may have precluded us from attaining statistical significance for high mitotic score and comedonecrosis.

Separation of patients according to their IDC-P risk status would ensure more adequate treatment for patients with IDC-P. Indeed, we recently showed that adjuvant radiotherapy could be beneficial for men with IDC-P [38]. However, radiation can cause serious side effects including urinary, bowel, and rectal complications as well as erectile dysfunction [39]. Our proposed stratification could help avoid overtreatment and spare men with low-risk IDC-P toxicity from an unnecessary treatment. The same applies to cases where IDC-P diagnosis is uncertain and in which PTEN loss and ERG expression are not observed [12,32]. Our criteria could serve as a prognostic tool to isolate patients at higher risk of recurrence. Moreover, our criteria could be easily implemented in the clinical setting. Reviewing every slide in detail with IDC-P would not be clinically feasible. Using our approach, pathologists would only need to evaluate one slide with the highest amount of IDC-P. Furthermore, for three of the four criteria (CINC, vessels, and comedonecrosis), reporting their presence or absence is sufficient. As for assessing the mitotic score, it is already routinely used in the grading of breast cancer [40] and sarcomas [41] in clinical laboratories and could also be applied toward IDC-P assessment. 

One of the strengths of this study is that we were able to validate our adverse criteria in an independent cohort. Additionally, we obtained fair to moderate interobserver agreement (overall κ 0.47, range 0.33–0.60) for the four proposed criteria. The strongest agreement (almost substantial) was seen for the assessment of mitotic score, indicating good reproducibility. Unsurprisingly, CINC and vessels showed the weakest agreement. The evaluation of these two criteria still requires a thorough examination of all the IDC-P lesions since, when scarce, their presence can be easily missed. Agreement for comedonecrosis, which can be diagnosed at lower magnification, was slightly better. Notably, these κ values are similar to the overall interobserver agreement obtained between 41 pathologists for the assigned Gleason score groups on 38 prostate biopsies (κ 0.44, range 0.00–0.88) [42]. We therefore believe that the proposed criteria are likely to be clinically applicable after thorough external validation. If we are unable to obtain acceptable levels of interobserver agreement with some of the criteria during external validation, we could use machine learning to better classify each feature. 

The present study has some limitations. While our current data suggest that high mitotic score and comedonecrosis may not significantly add to our BCR prediction models, we believe that their individual impact will be better demonstrated in larger cohorts, especially since most adverse criteria were not overlapping in patients. Indeed, although not significant, the HR associated with the presence of one to four criteria increased with the number of adverse criteria included in the model. The same goes for our finding that the presence of any adverse criteria rather than the number of adverse criteria had the strongest effect on BCR. Regardless of the combination, more than 75% of patients with adverse criteria in our validation cohort had only one adverse criterion. We believe that the effect of the number of criteria on BCR will be more apparent in larger cohorts. Moreover, because of the small number of prostate cancer-specific death in our validation cohort (*n* = 9), we were not able to assess disease-specific survival. Further studies using larger cohorts will allow us to build multivariate models including all standard prognostic factors of PCa and to evaluate the association of the proposed criteria with disease-specific survival. Additionally, as the study of RP specimens ensures a better characterization of IDC-P, our results will also need to be validated in biopsies. Furthermore, we only evaluated criteria on H&E or HPS stained slides, but we could use specific markers such as CD34 [43,44], Ki67, and pHH3 [45] to more accurately quantify blood vessels, proliferation, or mitoses. 

## 5. Conclusions

In conclusion, our data suggest that morphologic criteria can be used to distinguish patients with lower-risk IDC-P from those with higher-risk IDC-P. Except for CINC, all criteria are typically related to tumor growth or size. We propose combining two to four criteria, whose presence are independent predictors of BCR, to stratify men with IDC-P according to their risk status. If confirmed in larger cohorts, current pathologic reporting should be reviewed to include the concept of low and high-risk IDC-P.

## Figures and Tables

**Figure 1 cancers-13-06243-f001:**
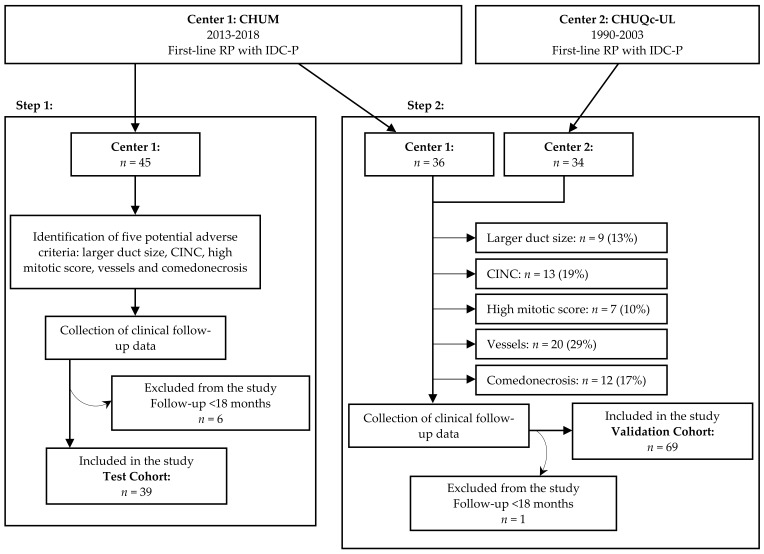
Study workflow. Adverse criteria were identified in patients from Center 1 (test cohort) then evaluated in an independent cohort of patients from Centers 1 and 2 (validation cohort). Centre hospitalier de l’Université de Montréal (CHUM) (Center 1); Centre de recherche du CHU de Québec-Université Laval (CHUQc-UL) (Center 2); RP: radical prostatectomy; IDC-P: intraductal carcinoma of prostate; CINC: cells with irregular nuclear contours.

**Figure 2 cancers-13-06243-f002:**
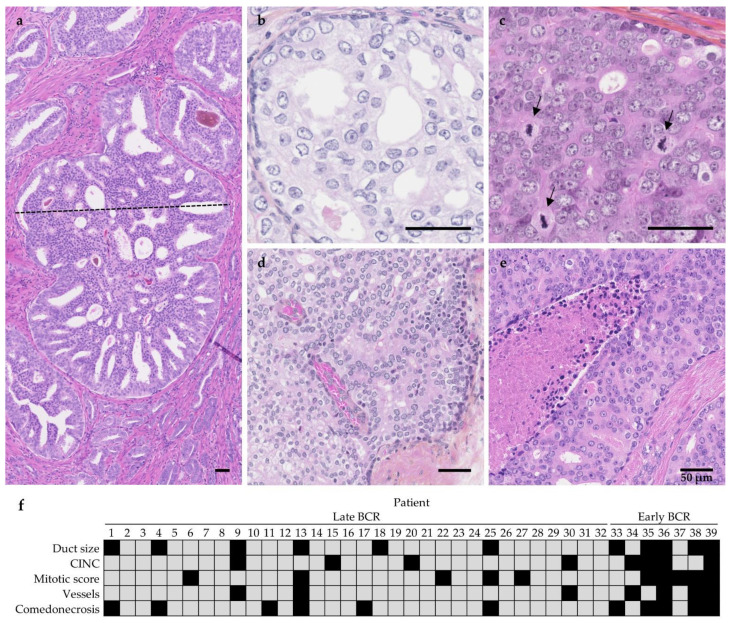
Proposed criteria to distinguish aggressive IDC-P. (**a**) Duct size calculated by measuring the transverse diameter (dotted line) of the largest duct, hematoxylin and eosin (H&E) staining. (**b**) Cells with irregular nuclear contours (CINC), hematoxylin phloxine saffron (HPS) staining. (**c**) Mitotic score obtained by counting the number of mitotic figures (arrows) in 10 high-power fields, HPS. (**d**) Blood vessels, HPS. (**e**) IDC-P comedonecrosis, H&E. Scale bars: 50 μm. (**f**) Distribution of the proposed criteria in the test cohort. Each column represents a patient and each line a criterion. Patients 1 to 32 were in the later BCR group while patients 33 to 39 were in the early BCR group. The presence of adverse criteria is represented by black squares.

**Figure 3 cancers-13-06243-f003:**
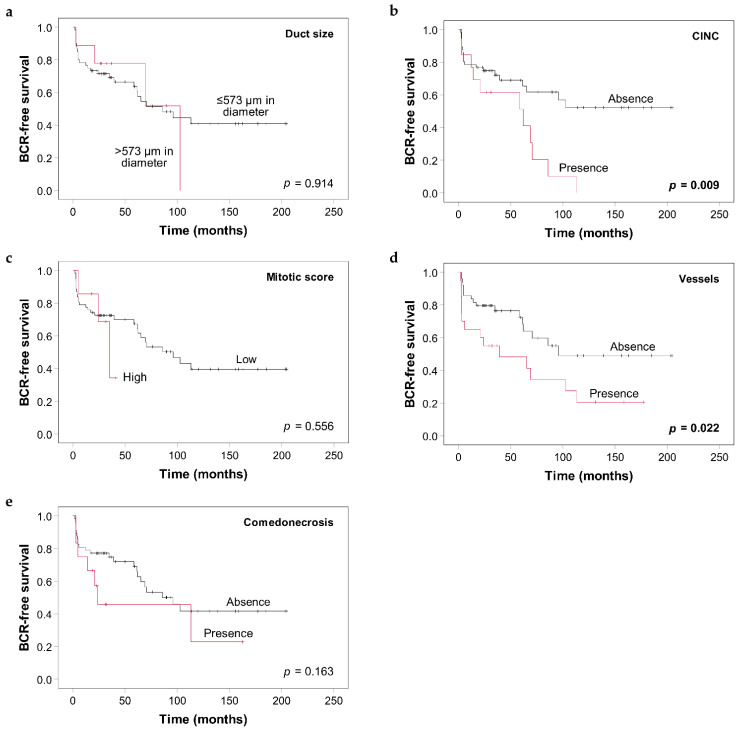
Kaplan–Meier curves of biochemical recurrence (BCR)-free survival according to the presence or absence of the five proposed criteria for IDC-P in the validation cohort: (**a**) Duct size, (**b**) CINC, (**c**) Mitotic score, (**d**) Blood vessels, and (**e**) Comedonecrosis. *p*-values were calculated using the log-rank test. CINC: cells with irregular nuclear contours.

**Figure 4 cancers-13-06243-f004:**
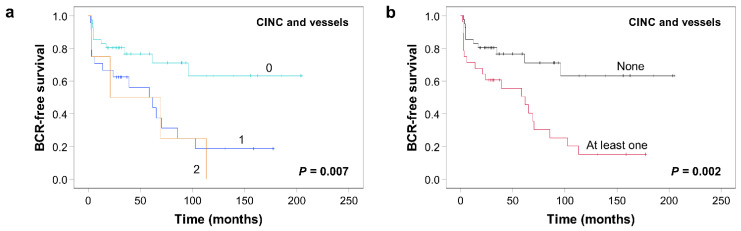
Kaplan–Meier curves of BCR-free survival according to the presence of CINC and the presence of vessels in IDC-P in the validation cohort. (**a**) Number of criteria. (**b**) Presence or absence of criteria. *p*-values were calculated using the log-rank test. CINC: cells with irregular nuclear contours.

**Table 1 cancers-13-06243-t001:** Clinicopathological characteristics of patients.

Characteristics	Test Cohort	Validation Cohort			*p*-Value	
	Center 1 *n* = 39	Center 1 *n* = 35	Center 2 *n* = 34	Total *n* = 69	Between Centers	Between Cohorts
Mean age at diagnostic (SD)	61.1 (5.3)	63.3 (5.6)	63.3 (7.2)	63.3 (6.4)	0.397 ^a^	0.054 ^a^
Mean pre-operative PSA (SD)	9.7 (6.9)	6.8 (4.2)	10.8 (9.5)	8.8 (7.5)	0.167 ^a^	0.556 ^a^
pT stage, *n* (%)					**0.003 ^b^**	0.099 ^b^
pT2	4 (10%)	6 (17%)	14 (41%)	20 (29%)		
pT3a	22 (56%)	16 (46%)	14 (41%)	30 (43%)		
pT3b	13 (33%)	13 (37%)	6 (18%)	19 (28%)		
Grade group, *n* (%)					**0.038 ^b^**	0.388 ^b^
1	0	0	5 (15%)	5 (7%)		
2	14 (36%)	14 (40%)	11 (32%)	25 (36%)		
3	13 (33%)	9 (26%)	13 (38%)	22 (32%)		
4	7 (18%)	2 (6%)	3 (9%)	5 (7%)		
5	5 (13%)	10 (29%)	2 (6%)	12 (17%)		
Lymphovascular invasion, *n* (%)	21 (54%)	15 (43%)	6 (18%)	21 (30%)	**0.001 ^c^**	**0.022 ^c^**
Positive margins, *n* (%)	12 (31%)	9 (26%)	8 (24%)	17 (25%)	0.708 ^c^	0.708 ^c^
Extraprostatic extension, *n* (%)	35 (90%)	29 (83%)	20 (59%)	49 (71%)	**<0.001 ^c^**	**0.012 ^c^**
Seminal vesicle invasion, *n* (%)	13 (33%)	13 (37.1%)	6 (18%)	19 (8%)	0.088 ^c^	0.650 ^c^
Biochemical recurrence, *n* (%)	15 (38%)	11 (31%)	20 (59%)	31 (45%)	**0.035 ^c^**	0.653 ^c^
Early biochemical recurrence, *n* (%)	7 (18%)	9 (26%)	8 (24%)	17 (25%)	1.000 ^c^	0.574 ^c^
Median follow-up in months (IQR)	54 (45–59)	35 (26–50)	157 (111–186)	67 (32–157)	**<0.001 ^b^**	0.086 ^b^

SD: standard deviation; PSA: prostate-specific antigen; IQR: inter-quartile range. Bold entities indicate statistically significant *p* values. ^a^ Welch’s *t*-test; ^b^ Mann–Whitney *U* test; ^c^ Pearson’s chi-square.

**Table 2 cancers-13-06243-t002:** Univariate Cox regression analysis for the prediction of BCR in the validation cohort.

Variables	Validation Cohort (*n* = 69)		
	HR	95% CI	*p*-Value
Mean pre-operative PSA	1.03	1.00–1.06	0.052
pT stage			
pT2	ref		
pT3a	1.71	0.68–4.32	0.257
pT3b	2.68	1.03–7.03	**0.045**
Grade group *			
1–2	ref		
3	2.72	1.07–6.92	**0.036**
4–5	5.05	1.96–13.02	**0.001**
Positive margins	1.72	0.79–3.75	0.176
Larger duct size	1.06	0.37–3.04	0.914
Presence of CINC	2.60	1.24–5.47	**0.012**
High mitotic score	1.44	0.42–4.91	0.560
Presence of vessels	2.24	1.10–4.55	**0.026**
Presence of comedonecrosis	1.81	0.77–4.24	0.117

PSA: prostate-specific antigen; CINC: cells with irregular nuclear contours; HR: hazard ratio; CI: confidence interval. * Grade groups 1–2 and 4–5 were combined because of the small number of patients in grade groups 1 (*n* = 5) and 4 (*n* = 5). Bold entities indicate statistically significant *p*-values.

**Table 3 cancers-13-06243-t003:** Cox regression analysis for the prediction of BCR in the validation cohort according to the presence of CINC and vessels in IDC-P.

Included Criteria	Variables	Validation Cohort (*n* = 69)					
		Univariate Analysis			Multivariate Analysis		
		HR	95% CI	*p*-Value	HR	95% CI	*p*-Value
CINC and vessels	Grade group *						
	1–2	ref			ref		
	3	2.72	1.07–6.92	**0.036**	2.22	0.86–5.74	0.101
	4–5	5.05	1.96–13.02	**0.001**	3.96	1.50–10.47	**0.006**
	Number of adverse criteria						
	0	ref					
	1	2.89	1.34–6.24	**0.007**			
	2	3.66	1.16–11.55	**0.027**			
	0 vs. ≥1	3.02	1.45–6.31	**0.003**	2.32	1.09–4.96	**0.029**

HR: hazard ratio; CI: confidence interval; CINC: cells with irregular nuclear contours. * Grade groups 1–2 and 4–5 were combined because of the small number of patients in grade groups 1 (*n* = 5) and 4 (*n* = 5). Bold entities indicate statistically significant *p*-values.

**Table 4 cancers-13-06243-t004:** Interobserver agreement between two observers for criteria assessment.

Criteria	Agreement, *n* (%)	Kappa (κ) ^a^	95% CI
CINC	79/108 (73)	0.33	0.14–0.51
Mitotic score	96/108 (89)	0.60	0.40–0.80
Vessels	86/108 (80)	0.47	0.28–0.66
Comedonecrosis	87/108 (81)	0.51	0.33–0.68
Overall	348/432 (81)	0.47	0.37–0.56

CI: confidence interval; CINC: cells with irregular nuclear contours. ^a^ Cohen’s kappa.

## Data Availability

The data presented in this study are available on request from the corresponding author. The data are not publicly available due to ethical restrictions.

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
