# Peer review of "Identification of Morphologic Criteria Associated with Biochemical Recurrence in Intraductal Carcinoma of the Prostate"

_cancers, 2021, doi:10.3390/cancers13246243_

Round 1

Reviewer 1 Report

The paper by Dr Diop et al deals with the identification of morphologic criteria associated with biochemical recurrence in intraductal carcinoma of the prostate (IDC-P). The authors found that IDC-P can be classified as low versus high-risk of biochemical recurrence.

The paper is of interest to both pathologists and clinicians.

Comments:

  • The authors should be more precise in the Summary when referring to: presence of larger duct size, cells with irregular nuclear contours (CINC) and high mitotic score. In particular, the actual size measurement of the ducts (as specified in the text) and the threshold between low and high mitotic score.
  • Blood vessels. Are the authors referring to small vessels in the stroma of intraductal papillary structures?
  • In the Introduction section, the authors might mention that a very small number of IDC-P cases are not associated with an invasive component.
  • Do the authors have any information on the clinical value of their findings when applied to prostate biopsies?
  • Haver the authors also explored to clinical significance of the number of ducts and acini involved?
  • Grade group (PG) 1. It is difficult to think that PG 1 can be associated with IDC-P
  • How often an invasive cribriform component was associated with IDC-P?

Reviewer 2 Report

Well written.

The manuscript is summarized from the principle of treatment to actual clinical practice.

In addition, it is described in detail and very easy to read.

In the future, it could be a fundamental indicator in this area.

Reviewer 3 Report

OK~ My comments and questions are as follows:

1) CINC, blood vessel features are very subjective, which can be controversial between observers. Given machine learning to classify each feature, they could be better and less arbitrary. Blood vessel in cribriform cancer cells in IDC-P can be seen in tangential sections, which could not be intratumoral components.

2) Quantity of measurement in CINC is very laborious and difficult in consensus. All of them should be compared with image analyzer. Since both comparison were missing, the data in two observers could be short of confidence.

Round 2

Reviewer 3 Report

In the next process, selected features can be meaningful. Confidential data-driven results remain still missing, but currently others look fair.  
